# CDC20 Holds Novel Regulation Mechanism in RPA1 during Different Stages of DNA Damage to Induce Radio-Chemoresistance

**DOI:** 10.3390/ijms25158383

**Published:** 2024-08-01

**Authors:** Yang Gao, Pengbo Wen, Chenran Shao, Cheng Ye, Yuji Chen, Junyu You, Zhongjing Su

**Affiliations:** 1Department of Histology and Embryology, Shantou University Medical College, Shantou 515041, China; gaoyang@stu.edu.cn (Y.G.); scr9711@163.com (C.S.); 21cye@stu.edu.cn (C.Y.); 21yjchen4@stu.edu.cn (Y.C.); 21jyyou@stu.edu.cn (J.Y.); 2School of Medical Information and Engineering, Xuzhou Medical University, Xuzhou 221002, China; wen_pengbo@xzhmu.edu.cn

**Keywords:** CDC20, RPA1, DNA damage repair, tumoral cell radio-chemosensitivity

## Abstract

Targeting CDC20 can enhance the radiosensitivity of tumor cells, but the function and mechanism of CDC20 on DNA damage repair response remains vague. To examine that issue, tumor cell lines, including KYSE200, KYSE450, and HCT116, were utilized to detect the expression, function, and underlying mechanism of CDC20 in radio-chemoresistance. Western blot and immunofluorescence staining were employed to confirm CDC20 expression and location, and radiation could upregulate the expression of CDC20 in the cell nucleus. The homologous recombination (HR) and non-homologous end joining (NHEJ) reporter gene systems were utilized to explore the impact of CDC20 on DNA damage repair, indicating that CDC20 could promote HR repair and radio/chemo-resistance. In the early stages of DNA damage, CDC20 stabilizes the RPA1 protein through protein-protein interactions, activating the ATR-mediated signaling cascade, thereby aiding in genomic repair. In the later stages, CDC20 assists in the subsequent steps of damage repair by the ubiquitin-mediated degradation of RPA1. CCK-8 and colony formation assay were used to detect the function of CDC20 in cell vitality and proliferation, and targeting CDC20 can exacerbate the increase in DNA damage levels caused by cisplatin or etoposide. A tumor xenograft model was conducted in BALB/c-nu/nu mice to confirm the function of CDC20 in vivo, confirming the in vitro results. In conclusion, this study provides further validation of the potential clinical significance of CDC20 as a strategy to overcome radio-chemoresistance via uncovering a novel role of CDC20 in regulating RPA1 during DNA damage repair.

## 1. Introduction

The DNA damage response is the main basis for tumor radiotherapy, as radiation-induced DNA double-strand breaks (DSB) act as the primary mechanism for inhibiting cell proliferation [1]. The radioresistance of tumor tissue is attributed to the abnormal enhancement of the DNA damage repair capacity [2]. According to the International Agency for Research on Cancer (IARC), cancer is one of the leading causes of death worldwide, with 19.96 million new cases and 9.74 million deaths reported in 2022 [3]. By 2050, the annual number of new cancer cases is projected to exceed 35 million, a 77% increase from 2022, underscoring the urgent need for effective cancer treatments. The literature suggests that tumor cells initiate DNA damage repair mechanisms upon exposure to radiation, ultimately determining the final fate of tumor cells (survival or death) [4,5]. Cellular protein machinery processes can lead to DNA degradation and can serve as molecular targets to combat cancer diseases [6,7]. Notably, recent studies have shed light on specific factors contributing to radiotherapy resistance in different types of cancer. For instance, the sustained activation of Poly(ADP-Ribose) Polymerase 1 (PARP1) in tumor cells with low epithelial cell transforming 2 (ECT2) expression inhibits nucleolar transcription, thereby promoting ribosomal DNA damage repair and causing radiotherapy resistance in lung cancer and nasopharyngeal carcinoma [8]. Similarly, elevated purine levels in glioma lead to radiotherapy resistance through DNA repair pathways [9]. Therefore, inhibiting the DNA damage repair ability of tumor cells holds potential as an effective way to overcome radiation resistance.

Cell division cycle 20 (CDC20) is a homolog of the cell division cycle 20 protein of Saccharomyces cerevisiae and is mainly involved in regulating the cell cycle and apoptosis processes. Increasing evidence shows that CDC20 is widely involved in the occurrence and development of tumors as an oncogene [10]. In our previous studies, we found that targeting CDC20 could regulate DNA damage repair and the radiation-induced endogenous apoptosis pathway through the Mcl-1/p-Chk1 signaling axis, thus affecting the radiosensitivity of colorectal cancer cells [11]. In this study, we aim to delve deeper into the mechanisms by which CDC20 regulates the resistance of cancer cells to radiotherapy and chemotherapy. As CDC20 is a key activator in the Anaphase-Promoting Complex/Cyclosome (APC/C), an E3 ubiquitin ligase [10], it remains unclear whether CDC20 influences DNA damage repair via ubiquitination mechanisms and thus affects drug resistance in tumor cells.

Replication Protein A (RPA) is a highly conserved heterotrimeric complex involved in various DNA processes such as replication, recombination, and repair. Previous studies have demonstrated that the presence of RPA1 in HEK293T cells significantly accelerates DNA breaks rejoining following radiation exposure, indicating its close association with DNA damage repair [12]. In this study, we aim to investigate whether CDC20 plays a role in regulating DNA damage repair by modulating the ubiquitination modification of RPA1 to gain deeper insights into the regulatory pathways involved in this critical cellular process.

In this study, we found that CDC20 accumulated massively in the nucleus after radiation exposure and regulated the homologous recombination repair process by regulating the stability of the RPA1 protein in response to radiation stimulation. This study further explores and expands upon the mechanisms by which CDC20 regulates DNA damage repair, thereby solidifying the potential of targeting CDC20 to inhibit abnormal DNA damage repair and overcome radiotherapy resistance. These findings contribute to a deeper understanding of the role of CDC20 in tumor treatment and provide a robust foundation for the development of CDC20-based therapeutic strategies.

## 2. Results

### 2.1. Radiation Induces Nuclear Accumulation of CDC20 and Affects DNA Damage Response

First, the expression levels of CDC20 after radiation were analyzed in eight cell lines from different cancer types, including KYSE70, KYSE200, KYSE450, H1299, TE10, HCT116, MDA-MB-231, and BT549. The results showed that both mRNA and protein levels of CDC20 were higher in all cells following irradiation (IR) compared to non-IR conditions (Figure 1A,B). Meanwhile, the nuclear accumulation of CDC20 was also triggered by IR. Immunofluorescence results showed that CDC20 appeared to have nuclear aggregation after IR exposure (Figure 1C). Additionally, the nucleocytoplasmic separation experiment further confirmed that the expression of CDC20 in the nucleus increased sharply (Figure 1D). The above results indicate that CDC20 may be involved in the IR stress response in the nucleus.

In response to the high incidence of esophageal cancer in the Chaoshan region of Guangdong Province, tumor cells, primarily KYSE200 and KYSE450, were selected for subsequent experiments. Additionally, HCT116 cells were also selected to investigate the potential role and mechanism of CDC20 in pan-cancer. To explore the relationship between CDC20 and DNA damage in cancer cells following IR exposure, RNA interference was employed to knock down CDC20 expression in KYSE200, KYSE450, and HCT116 cells. In response to DNA damage, the phosphorylation of H2AX at Ser139 (known as γH2AX) occurs, and the levels of γH2AX have been widely used as a sensitivity marker for DSBs [13]. The results revealed that CDC20 knockdown cells exhibited significantly higher γH2AX levels compared to control cells post-IR (Figure 1E,F). Given that CDC20 has been recognized as a crucial positive regulator of radiosensitivity in numerous tumors [14,15], and our findings demonstrated the increased nuclear accumulation of CDC20 following IR exposure, we hypothesized that the role of CDC20 in radioresistance may be linked to this phenomenon.

### 2.2. CDC20 Regulates the Homologous Repair of DSBs

Since nonhomologous end-joining (NHEJ) and homologous recombination (HR) are identified as key pathways that are involved in repairing DSBs in eukaryotic cells [16], the HR and NHEJ reporter gene systems were utilized to explore the impact of CDC20 on DNA damage repair (Figure 2A,B). As shown in Figure 2C,D, the knockdown of CDC20 slightly inhibited NHEJ activity, but this effect was not statistically significant. Conversely, the knockdown of CDC20 resulted in a significant decrease in HR activity. Next, the expression levels of classic proteins, Rad51 in the HR pathway and Ku70/Ku80 in the NHEJ pathway, were detected [17]. As shown in Figure 2E,F, the knockdown of CDC20 in KYSE200 and KYSE450 cells resulted in a significant reduction in RAD51 protein expression. However, the protein expression levels of Ku70 and Ku80 remained largely unaltered (Figure 2E,F). Collectively, nuclear accumulation of CDC20 after radiation plays an important role in the HR repair process in response to DNA damage.

### 2.3. CDC20 Interacts with RPA1/RPA2 to Assist DNA Damage Repair

To determine the association of CDC20 with other factors involved in DNA damage repair upon nuclear entry, coimmunoprecipitation (co-IP) was conducted in KYSE450 cells using CDC20 antibodies, followed by liquid chromatography with tandem mass spectrometry (LC-MS/MS). Figure 3A shows the results of Coomassie brilliant blue staining and Western blot verification. LC-MS/MS data showed that CDC20 exhibited interaction with 534 proteins (Figure 3B). Subsequent functional analysis of those proteins was performed using the online database Metascape, which categorized them based on coverage scores, resulting in the identification of a subset of 28 proteins commonly involved in the DNA damage repair pathway (Figure 3C). Among them, RPA1 and RPA2 are strongly associated with multiple DNA repair processes, particularly HR [18,19]. As shown in Figure 3D, CDC20 was redistributed after IR treatment and co-localized with RPA1/RPA2 foci in the nucleus. To confirm this result, a co-IP assay was performed on three cell lines. The results showed that IR exposure significantly increased the amount of RPA1/RPA2 that binds with CDC20 (Figure 3E). The above results suggest an interaction between CDC20 and RPA1/RPA2 in relation to DNA repair processes.

### 2.4. CDC20 Mediates the Timely Unloading of RPA1 during DNA Damage Repair

During the early stages of DNA damage, RPA1 and RPA2 proteins are recruited to participate in the repair process [20]. As the repair process progresses, RPA from the damaged DNA region needs to be unloaded to allow subsequent proteins, such as Rad51, to perform their functions. Rad51 is a key protein involved in HR that repairs DNA DSBs by utilizing an undamaged sister chromatid as a template [20,21].

To gain more insight into the CDC20 function in HR, the protein expression level of RPA1/RPA2 was examined. Aht 24 h after treatment with 5 Gy X-ray, RPA1 and RPA2 protein levels were upregulated in CDC20 knockdown cells (Figure 4A,B). This causal relationship between the decrease in CDC20 expression and the subsequent increase in RPA1/RPA2 expression remains vague, despite the potential involvement of CDC20 in positively regulating the stability of RPA1/RPA2. Since CDC20 is a key activator in the Anaphase-Promoting Complex/Cyclosome (APC/C), an E3 ubiquitin ligase [10], we therefore tested if CDC20 affects RPA1 ubiquitination. Indeed, there was a decrease in RPA1 ubiquitination in CDC20 knockdown cells as compared to control cells (Figure 4C). This indicated that CDC20 is involved in regulating the ubiquitination of RPA1. Furthermore, levels of endogenous RPA1 protein were thus examined in CDC20 knockdown cells in the presence of cycloheximide (CHX), an inhibitor of protein biosynthesis in eukaryotic organisms. As seen in Figure 4D, the half-life of the RPA1 protein in CDC20 knockdown cells was significantly extended compared to that in control cells.

Collectively, we determined that CDC20 is involved in the later stage of the DNA damage repair process by promoting the ubiquitination of RPA1.

### 2.5. CDC20 Regulates Radio- and Chemosensitivity in Cancer Cells

To further investigate the role of CDC20 in cancer cells, CCK-8 was utilized to evaluate the effect of CDC20 on the viability of cells. The results indicated that the knockdown of CDC20 significantly decreased cell viability upon radiation (Figure 5A). Consistently, similar results were obtained in a colony formation assay (Figure 5B). Furthermore, apcin, as a specific CDC20 inhibitor, was used to further evaluate the role of CDC20 in radiosensitivity. As shown in Figure 5C, apcin pretreatment plus X-ray radiation had significantly greater inhibitory effects on cell viability than radiation exposure alone.

After DNA damage, the ataxia-telangiectasia-mutated-and-Rad3-related kinase (ATR) protein is recruited and activated, initiating the downstream phosphorylation of checkpoint kinase 1 (CHK1). This activation of CHK1 enables a temporary delay in cell cycle progression, allowing for DNA damage repair [22]. Further results confirmed that CDC20 knockdown reduced ATR-mediated CHK1 phosphorylation and increased the phosphorylation level of γH2AX (Figure 5D). These results implied that CDC20 regulates radiosensitivity in cancer cells by the ATR-CHK1 pathway.

Next, we explored whether CDC20 has a potential effect on the chemosensitivity of cancer cells. Cisplatin and etoposide, two commonly used chemotherapy agents known to induce DNA damage in cancer cells, were employed. The results consistently supported our hypothesis that CDC20 knockdown could sensitize cancer cells to both cisplatin and etoposide in a dose-dependent manner (Figure 5E,F). Subsequently, as shown in Figure 5G,H, the expression of γH2AX was significantly increased following both cisplatin and etoposide treatment combined with CDC20 knockdown in KYSE450 cells. Similarly, the phosphorylation levels of CHK1 decreased in CDC20 knockdown cells (Figure 5G,H). In addition, even when using an appropriate concentration (20 μM) of cisplatin or etoposide to treat cancer cells, higher levels of γH2AX protein could still be detected in CDC20 knockdown cells (Figure 5I,J). These findings suggest that CDC20 is also involved in the development of chemoresistance by promoting higher DSBs.

### 2.6. Apcin Regulates Radiosensitivity In Vivo

To determine whether CDC20 inhibitors could enhance radiosensitivity in vivo, we conducted a tumor xenografth model in athymic nude mice. The tumors were allowed to grow until they reached a size of approximately 100 mm^3^. Then, the mice were randomly divided into three treatment groups (Figure 6A). As shown in Figure 6B, both IR alone and apcin alone moderately inhibited the growth of xenograft tumors generated from KYSE450 cells, whereas significant inhibition was observed when IR was combined with apcin. The tumor weight and volume in the apcin combined with the IR group were markedly lower than those in the other treatment groups (Figure 6C,D), indicating that inhibition of CDC20 activity increased radiosensitivity in vivo.

In addition, TUNEL staining was performed to detect cell apoptosis in tumor tissues. As shown in Figure 6E,F, both IR alone and apcin alone slightly promoted tumor cell apoptosis, whereas a more significant effect was observed when IR was combined with apcin. This indicates that the combination of IR and apcin enhances radiation-induced apoptosis in KYSE450 xenografts. Furthermore, H&E staining results showed distinct differences between the groups. In the control (0 Gy) group, cells were tightly arranged and actively dividing, and blood vessels were dense. In contrast, the Apcin + 10 Gy group exhibited necrotic and vacuolated cells, infiltration of inflammatory cells (mainly lymphocytes), increased interstitial components, and increased fibrous bundles. These observations suggest impaired cell proliferation ability in the combination treatment group (Figure 6G). Moreover, immunohistochemical (IHC) staining showed that IR combined with apcin clearly increased the expression of γH2AX in tumors (Figure 6H). Collectively, these results indicate that inhibition of CDC20 activity increases radiosensitivity in vivo by suppressing DNA damage repair.

## 3. Discussion

Targeting the DNA damage response (DDR) signaling pathway has emerged as a promising strategy to overcome radiation resistance in cancer treatment [2]. Significantly, remarkable progress and breakthroughs have been achieved in recent years through the development of targeted drugs that inhibit DDR enzymes, such as PARPi [23]. Nonetheless, it is imperative to acknowledge the emergence of PARPi resistance as a substantial challenge in the field of clinical cancer treatment [24]. Consequently, there is an urgent need for the development of novel targets for the DDR pathway.

When radiation or chemical-induced DNA damage occurs, specific proteins, such as DNA repair enzymes, signaling molecules, and transcription factors, are activated and translocated into the nucleus [25]. Their nuclear localization is essential for the regulation of the DDR process. In our study, we illuminate a novel mechanism of CDC20 in response to DNA damage, particularly its nuclear accumulation post-radiation exposure. This accumulation suggests an active role for CDC20 in the DDR. Following ionizing radiation-induced DNA DSB, a cascade of cellular responses is triggered, encompassing damage detection, repair, and DNA synthesis [26]. This process involves multiple pathways, with HR and NHEJ representing the predominant mechanisms for DSB repair [16]. In recent years, research has elucidated that proteins within non-DNA damage response pathways may potentially engage directly or indirectly in DNA damage responses [27,28,29]. Our previous study demonstrated that targeting CDC20 could downregulate the expression of MCL-1/RAD51 [11]. In this current study, we further observed that CDC20 primarily influences the HR pathway. These findings contribute to a deeper understanding of the multifaceted functions of CDC20 in DDR.

This study demonstrated a novel regulation mechanism of CDC20 in DNA repair involving RPA1. In the early stages of HR, single-stranded DNA (ssDNA) generated by end resection is rapidly encapsulated by the single-strand binding protein RPA [30,31]. This process ensures the polarity of DNA end resection while safeguarding ssDNA from degradation by other nucleases. It prevents the self-formation of secondary structures in ssDNA, facilitating the activation of DDR through checkpoint kinase activation [32]. In our current study, we made an interesting observation concerning the early stages of DNA damage. We found that the binding of CDC20 to its target proteins did not induce their degradation as expected; instead, it appeared to enhance the stability of RPA1/2. Previous studies have reported that HSF1 inhibits the activity of APC/C by binding to CDC20, which subsequently results in the failure of mitotic exit [33]. This implies the existence of other co-regulatory factors that may modify the function of the E3 ubiquitin ligase (CDC20) towards its target protein (RPA1). Small evidence is available regarding the interaction between CDC20 and RPA1. A study in hepatocellular carcinoma showed that CDC20 and Rfc4 are involved in cancer cell survival [34], and another study showed that Rfc4 interacts with RPA1 for DNA replication and DNA damage repair [35]. The SUMOylation of RPA1 regulates the affinity between RPA1 and RAD51 to promote HR-mediated repair of DSBs [36]. Additional studies are necessary to unravel the interactions between CDC20 and RPA1.

While it is known that RPA can tightly bind to ssDNA, the presence of RPA impedes the subsequent steps of HR. Therefore, in the later stages of damage, RPA must be replaced by the recombinase Rad51 [37]. It has been shown that downregulating the expression of the CDC20 protein and blocking the subsequent binding of RAD51 to DNA damage sites can effectively reverse radiotherapy resistance [11]. Inhibiting RPA1 appears to increase the radiosensitivity of esophageal cancer cells [19,38]. In our current study, we observed that CDC20 can ubiquitinate and degrade RPA1 in the later stage of DDR. Furthermore, considering the findings from previous reports, we speculate that the degradation of RPA1 by CDC20 might promote the binding of RAD51 to damaged DNA sites. This could facilitate efficient DNA repair processes.

Our study reports a novel “two-phase” function of CDC20 in DNA damage repair. Once DNA damage occurs, the DDR orchestrates a coordinated and intricate cellular reaction by activating two parallel pathways: (1) The DNA damage checkpoint pathway, which induces cell cycle arrest at the G2/M phase, inhibits chromosome separation and mitotic exit until the damaged DNA is successfully repaired. (2) The activation of a suite of DNA repair factors that collaborate to mend the broken DNA strands [39]. Upon completion of DDR, the DNA damage checkpoint must be deactivated, allowing the cell to exit the arrested G2/M phase and resume normal cell cycle progression [40]. It is worth noting that previous studies have reported the role of CDC20 in chromosome separation, mitotic exit, and DNA damage repair separately [11,41]. Our study reveals the two-phase functionality of CDC20 in the early and late stages of DNA damage, which further elucidates the complexity of CDC20’s function and its potential as a viable therapeutic target. Additional studies are necessary to explore the role of CDC20 in cell cycle regulation and its association with DNA damage repair. In particular, the mechanism by which CDC20 plays a regulatory role between cell cycle regulation and DNA damage repair should be explored. This paper focuses on how CDC20 regulates the activity of RPA1/RPA2 and its impact on DNA damage repair pathways, as well as how CDC20 regulates the deloading process of RPA1, but the exact mechanisms linking CDC20 and RPA1 require research.

CDC20 is frequently overexpressed in various malignant tumors and exerts its main function by promoting resistance to both chemotherapy and radiation [11,42,43]. CDC20 activates the anaphase-promoting complex (APC) to form the E3 ubiquitin ligase complex APCCDC20, which promotes mitosis. CDC20 ensures adequate chromosome segregation. CDC20 is overexpressed in many cancers and has been identified as a potential treatment target [44]. This study investigated and enhanced our understanding of the molecular mechanisms underlying CDC20’s role in cell cycle control and DDR, thereby reinforcing its potential as a promising therapeutic target for malignant tumors. Hence, CDC20 inhibitors may be a novel therapeutic strategy for the treatment of radiotherapy and chemoresistance. However, it is worth noting that CDC20 also plays a critical role in the regulation of the normal cell cycle. Consequently, targeting CDC20 may potentially result in cell cycle disturbances in normal tissues [45]. Several CDC20 inhibitors are known (e.g., apcin, diosgenin, TAME, and CFM-4, among others) [44], but so far they have only been assessed in vitro or, on rare occasions, in vivo. Their actual clinical efficacy and safety are unknown, and no clinical trials of CDC20 inhibitors are currently underway. Therefore, exploring the efficacy and safety of CDC20 inhibitors in the treatment of tumors, as well as the possibility of the combination of CDC20 inhibition with other therapeutic targets, may represent a promising avenue for future interventions in the treatment of malignant tumors.

## 4. Materials and Methods

### 4.1. Cell Culture and Radiation

Esophageal squamous cell carcinoma (ESCC) cell lines (KYSE70, KYSE200, KYSE450, TE10), breast cancer cell lines (MDA-MB-231, BT549), a colorectal cancer cell line (HCT116), and a non-small cell lung cancer (NSCLC) cell line (H1299) were utilized in this study.

The human cell lines KYSE200, KYSE450, MDA-MB-231, and BT549 cells were kindly provided by Dr. Liyan Xu (Shantou University Medical College [STU], Shantou, China) [46,47]. KYSE70 was kindly provided by Dr. Enmin Li (STU) [48]. TE10 was obtained from Dr. Xu [49]. The human cell lines HCT116 and H1299 cells were kindly provided by Dr. Guoping Zhao (Hefei Institutes of Physical Science, Chinese Academy of Sciences, Hefei, China) [50,51]. KYSE70, KYSE200, and TE10 cells were maintained in RPMI-1640 medium (Sigma, Shanghai, China). KYSE450 and HCT116 were maintained in the DMEM medium (Biological Industries, Beit HaEmek, Israel). MDA-MB-231, BT549, and H1299 were maintained in DMEM/F12 medium (Biosharp, Hefei, China). All mediums were supplemented with 10% fetal bovine serum (Vivacell, Denzlingen, Germany) and 1% penicillin/streptomycin (Biosharp, Hefei, China).

As for the different treatments of cells, we utilized specific compounds obtained from Selleck (Shanghai, China), including Cycloheximide, Cisplatin, and Etoposide. Moreover, the CDC20 inhibitor, apcin, was purchased from Tocris, Inc. (Minneapolis, MN, USA). Furthermore, the irradiation was carried out using an X-ray irradiator, RS 2000 X-RAY Irradiator (Rad Source, Buford, GA, USA). After irradiation, cells were cultured at 37 °C for the indicated times.

### 4.2. qRT-PCR Analysis

Total RNA was prepared from cultured cells using Trizol (Invitrogen, Shanghai, China) according to the manufacturer’s protocol. The kit for reverse transcription PCR (RT-PCR) was purchased from GeneStar (Beijing, China). Amplification of the generated cDNA was carried out in SYBR qPCR Mix (GeneStar) with an ABI StepOneTM Real-Time PCR instrument (GeneStar). Each measurement was performed in triplicate, and the results were normalized by the expression of the U6 gene. Fold change relative to the mean value was determined by 2^−ΔΔCt^. The primers used for qRT-PCR analysis were purchased from General Biol (Chuzhou, China) and listed in Appendix A. Reaction parameters were 95 °C for 2 min, 95 °C for 5 s, and 60 °C for 10 s for 40 cycles.

### 4.3. Western Blot

The Western blot procedure was as described previously [52]. In brief, following the extraction of total cellular proteins, the proteins undergo concentration determination and denaturation. Subsequently, after electrophoresis and transfer, the polyvinylidene fluoride (PVDF) membrane carrying the loaded proteins is subjected to overnight incubation with primary antibodies (GAPDH (60004-1-Ig, 1:2000, Proteintech, Wuhan, China), CDC20 (10252-1-AP, 1:1000, Proteintech, Wuhan, China), γH2AX (sc-517348, 1:1000, Santa Cruz, CA, USA), Ku70 (A7330, 1:1000, Abclonal, Wuhan, China), Ku80 (A5862, 1:1000, Abclonal, Wuhan, China), Rad51 (8875S, 1:1000, Cell Signaling Technology [CST], Danvers, MA, USA), RPA1 (A3367, 1:1000, Abclonal, Wuhan, China), RPA2 (A2189, 1:1000, Abclonal, Wuhan, China), ATR (sc-515173, 1:1000, Santa Cruz, CA, USA), P-ATR (30632, 1:1000, CST, Danvers, MA, USA), Chk1 (25887-1-AP, 1:1000, Proteintech, Wuhan, China), p-Chk1 (2348S, 1:1000, CST, Danvers, MA, USA)). This is followed by additional incubation with secondary antibodies. Following completion, the membrane is subjected to machine visualization. Quantitative analysis of the protein bands is then carried out using ImageJ software (version number: 1.8.0).

### 4.4. Immunofluorescence Staining

Cells were fixed with 4% paraformaldehyde after a brief rinse with PBS. 0.2% Triton X-100 was used to permeabilize cells. Afterwards, cells were blocked for 2 h. Next, cells were incubated overnight at 4 °C with primary antibodies diluted in blocking buffer (CDC20 (10252-1-AP, 1:500, Proteintech, Wuhan, China), γH2AX (9718S, 1:500, CST, Danvers, MA, USA), RPA1 (NBP2-37500, 1:500, Novus, CO, USA), RPA2 (AB111161, 1:500, Abcam, Cambridge, UK)). On the second day, after washing the cells with PBS, incubate the secondary antibody for 1 h at room temperature in the dark. Nuclei were counterstained with DAPI.

### 4.5. Transfection of siRNA and shRNA Sequence

The siRNA (si-CDC20) and shRNA sequences were purchased from GenePharma (Shanghai, China), and siRNA transfection was carried out using Lipofectamine 2000 (Thermo Fisher, Carlsbad, CA, USA) according to the manufacturer’s protocols. The siRNA sequences and shRNA sequences are listed in Appendix A.

### 4.6. Coimmunoprecipitation (Co-IP)

Total cell lysates were prepared using RIPA buffer containing protease inhibitors. After centrifugation, the primary antibody (CDC20 (10252-1-AP, 1 μg, Proteintech, Wuhan, China)) was added to the supernatant and incubated at 4 °C for 12 h while gently stirring. Protein G Sepharose Bead Slurry (Santa Cruz, CA, USA) was then added to capture the protein complex. After incubation at 4 °C for 4 h with gentle agitation, the samples were centrifuged for 5 min at 4 °C. The supernatant was discarded, and the pellet was washed with RIPA buffer. Finally, the immunoprecipitates were resuspended using Western blot analysis using an SDS-PAGE loading buffer.

### 4.7. LC-MS/MS Assay

KYSE450 cells were treated with 5 Gy X-ray and harvested. Whole-cell protein was extracted using cell lysis buffer (Beyotime, Shanghai, China) for IP supplemented with PMSF (Beyotime, Shanghai, China) and immunoprecipitated with CDC20 antibody (10252-1-AP, 1 μg, Proteintech, Wuhan, China). The elution products were separated by SDS-PAGE, and the bands were cut from the gel as closely as possible. After the protein digestion, these extracts were analyzed by LC-MS/MS.

### 4.8. Cell Viability and Colony Formation Assay

The KYSE450 and KYSE200 cells were seeded at a density of 2 × 10^3^ per well in 96-well plates. At the indicated time post-X-ray treatment, cell viability was analyzed using the CCK8 kit (Biosharp, Hefei, China) according to the manufacturer’s instructions. The optical density (OD) value was measured using a microplate reader (Biotek, Winooski, VT, USA, model Elx800) at a wavelength of 450 nm.

In the colony formation test, a total of 800 cells were seeded in a 60-mm dish. After irradiation, the dishes were incubated for two weeks at 37 °C in a 5% CO_2_ incubator for 10 days. Then, the dishes were washed with PBS, fixed with a solution containing methanol: acetic acid (V/V = 9:1) for 30 min, and subsequently stained with crystal violet for 30 min. The colonies containing more than 50 cells per colony were scored and plotted.

### 4.9. Xenograft Model

Six-week-old male BALB/c-nu/nu mice were purchased from Hangzhou Ziyuan Laboratory Animal Technology Co. (Hangzhou, China). The mice were randomly divided into four groups, each containing three mice, and then stably transfected KYSE450 cells (5 × 10^6^) were inoculated subcutaneously in each mouse. On the eighth day post-injection, the first and second groups were exposed to normal light conditions, while the third and fourth groups received a single dose of 10 Gy irradiation. Starting from the ninth day post-injection, the second and fourth groups received intraperitoneal administration every other day (20 mg/kg Apcin), while the remaining groups were administered an equivalent volume of saline for 2 consecutive weeks. Tumor dimensions and volumes (mm^3^) were measured and calculated with calipers every three days. Finally, the nude mice were sacrificed using severed neck execution on the 29th day after inoculation, and the tumors were harvested. The tumor tissues were fixed in 4% paraformaldehyde to obtain paraffin sections for the TUNEL assay, H&E staining, and immunohistochemical analysis. All animal experimental procedures were approved by the Animal Care and Use Committee of Shantou University Medical College (SUMC2022-034).

### 4.10. IHC Analysis

Harvested tumors were fixed in 4% formalin buffer, embedded in paraffin blocks, and sectioned at 5 µm onto slides. Sections were immersed in boiling sodium citrate buffer (pH 6.0) for antigen retrieval. Next, sections were incubated with 3% H_2_O_2_ for 10 min to block endogenous peroxidase activity. Then, the slides were incubated with the first antibodies against γH2AX (CST, 9718S, 1:100) at 4 °C overnight. Next, the slides were incubated with a secondary antibody at 37 °C for 30 min. Immunoreactivity was detected using the Strept Avidin-Biotin Complex (SABC) method.

### 4.11. TUNEL Assay

For the TUNEL assay, tissue sections were fixed with 4% paraformaldehyde for 30 min at room temperature. After washing with PBS, cells were permeabilized with 0.1% Triton X-100 in 0.1% sodium citrate for 2 min on ice. The TUNEL reaction mixture (Roche, Shanghai, China) was then applied according to the manufacturer’s instructions. After incubation for 1 h at 37 °C in a humidified atmosphere in the dark, cells were rinsed three times with PBS. Nuclear DNA was counterstained with DAPI for 5 min. Fluorescence microscopy was used to evaluate apoptotic cells, which were identified by TUNEL-positive staining.

### 4.12. H&E Staining

For H&E staining, tissue sections were cut into 6 μm, dewaxed, rehydrated, and immersed in Mayer hematoxylin, stirring for 3–5 min. After rinsing with H_2_O, sections were stained with a 1% eosin Y solution with agitation for 1–3 min. Subsequently, the sections were dehydrated with alcohol and xylene. Finally, the mounting medium was added prior to covering with a cover slip.

### 4.13. Statistical Analysis

The statistical significance of differences between groups was assessed using the GraphPad Prismj8 software (San Diego, CA, USA). All data are presented as the mean ± SD of the mean unless otherwise specified. Student’s *t*-test or one-way ANOVA was used to determine the significance of all quantitative experiments. All experiments were repeated at least three times. *p*  <  0.05 was considered significant. Statistical significance was defined as * *p*  <  0.05, ** *p*  <  0.01, *** *p*  <  0.01, **** *p* < 0.0001, ns = not significant.

## 5. Conclusions

In summary, targeting CDC20 exacerbates the heightened levels of DNA damage induced by radiation or chemotherapy, thereby augmenting the sensitivity of tumor treatment. Notably, our study provides novel insights into the role of CDC20 in DNA damage response and repair, highlighting its interaction with RPA1 as a crucial factor in this process. These findings not only enhance our understanding of the molecular mechanisms underlying DNA repair but also open up new possibilities for targeted cancer therapies.

## Figures and Tables

**Figure 1 ijms-25-08383-f001:**
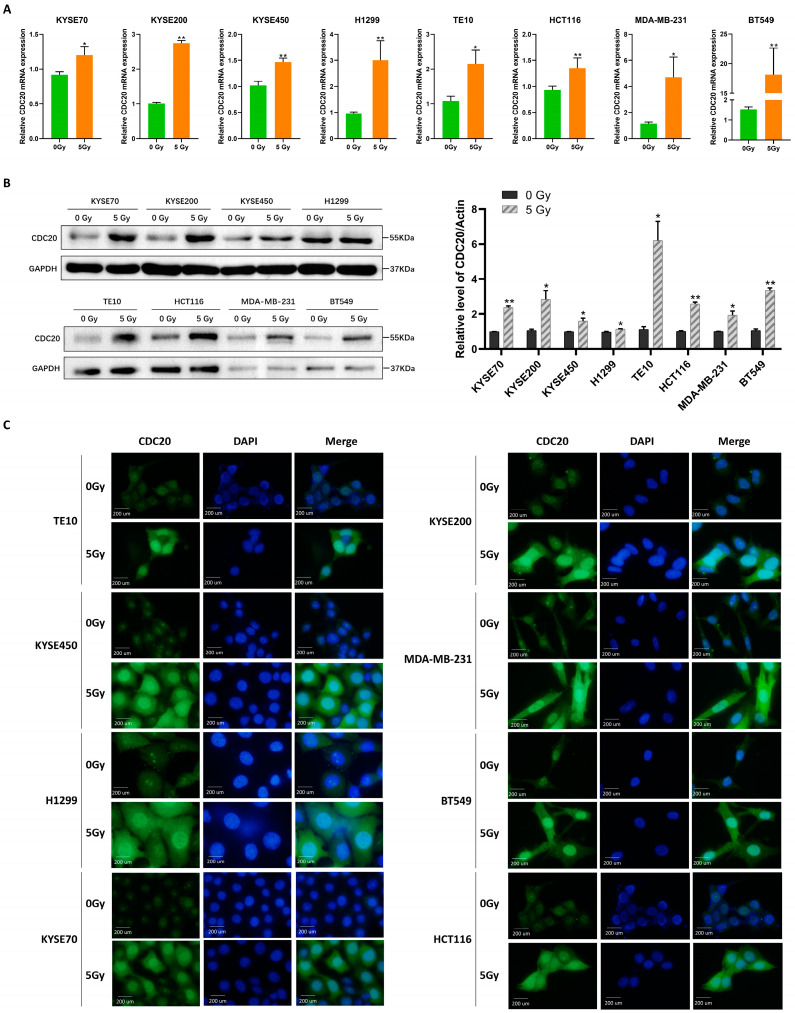
Radiation induces nuclear accumulation of CDC20 and affects DNA damage response. (**A**,**B**) Transcriptional (**A**) and translational levels (**B**) of CDC20 in KYSE70, KYSE200, KYSE450, H1299, TE10, HCT116, MDA-MB-231, and BT549 cells after 5 Gy X-ray were detected by quantitative real-time PCR and immunoblotting. (**C**) Immunofluorescence detection of CDC20 protein localization in TE10, KYSE450, H1299, KYSE200, MDA-MB-231, KYSE70, HCT116 and BT549 cells after radiation (magnification, ×100). The CDC20 protein is dyed green and the nucleus blue. (**D**) Nuclear and cytosolic lysates were prepared from KYSE200, KYSE70, KYSE450, H1299, TE10, HCT116, MDA-MB-231, and BT549 cells treated with 5 Gy X-ray and were subjected to immunoblotting analysis at 24 h. (**E**,**F**) The expression level of γH2AX was detected by Western blotting (**E**) and immunofluorescent staining (**F**) in CDC20 knockdown KYSE200 cells, KYSE450, and HCT116 cells compared with control shRNA cells upon 5 Gy X-ray after 24 h. GAPDH was used as an internal control. Data were pooled from three independent experiments, and the results were represented as mean ± SD. * *p* < 0.05, ** *p* < 0.01, *** *p* < 0.001, **** *p* < 0.0001, ns = not significant.

**Figure 2 ijms-25-08383-f002:**
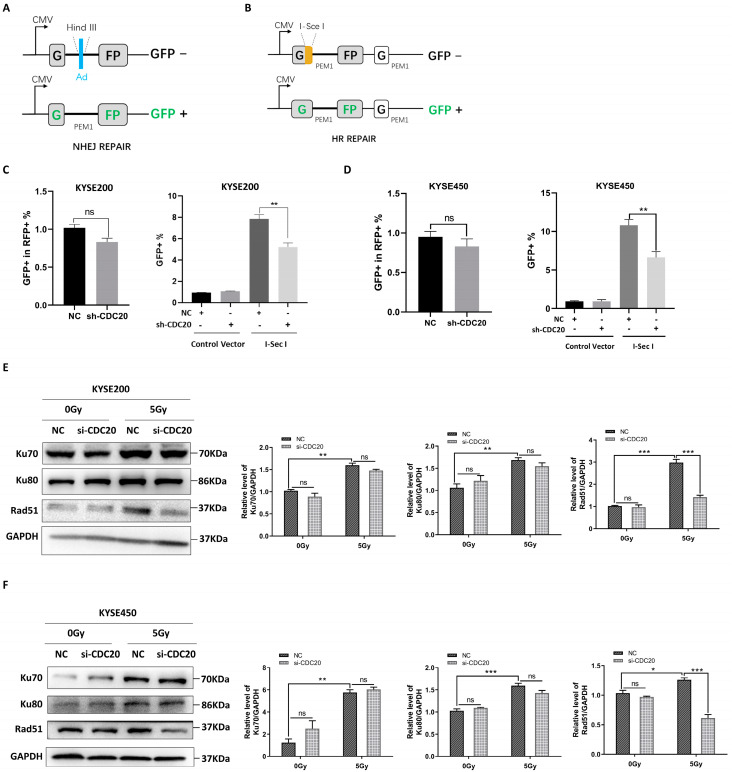
CDC20 regulates homologous repair of DSBs. (**A**,**B**) Reporter constructs for the analysis of NHEJ (**A**) and HR (**B**). (**C**,**D**) CDC20 knockdown cells ((**C**): KYSE200, (**D**): KYSE450) were co-transfected with 2 μg of pDsRed2-N1 plasmid (RFP) and 2 μg of HindIII-digested NHEJ reporter gene plasmid (GFP). 48 h after transfection, cells were harvested and used to analyze GFP-expressing and RFP-expressing cells. 3 μg of DR-GFP plasmid was transfected into knockdown CDC20 cells, and 24 h after transfection, 5 μg of I-Sce I expression plasmid was transferred into shCDC20 cells. After 24 h, homologous recombination induced by I-Sce I was detected, and the results were statistically analyzed. (**E**,**F**) After irradiating with 5 Gy X-ray, the expression level of Ku70, Ku80, and Rad51 were detected by immunoblotting analysis in CDC20 knockdown KYSE200 (**E**) and KYSE450 (**F**) cells compared with control vector cells after 24 h. GAPDH was used as an internal control. Data were pooled from three independent experiments, and the results were represented as mean ± SD. * *p* < 0.05, ** *p* < 0.01, *** *p* < 0.001, ns = not significant.

**Figure 3 ijms-25-08383-f003:**
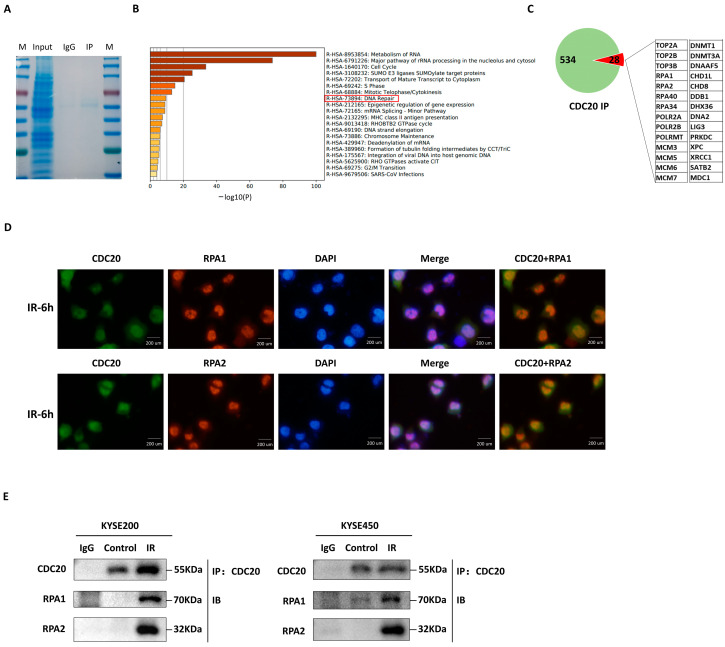
CDC20 interacts with RPA1/RPA2 to assist DDR. (**A**) Validation of CDC20 IP experiments with Coomassie Brilliant Blue and Western blotting. (**B**) Pathway enrichment analysis of CDC20-interacting proteins (red box indicates enrichment to DNA repair pathways). (**C**) The pie chart shows the proteins (534) that interact with CDC20, of which 28 are related to DDR. These 28 proteins are displayed in list form. (**D**) KYSE450 was irradiated with a 5 Gy X-ray, and its specific antibodies were used to detect the localization of CDC20, RPA1, and RPA2 6 h later. (**E**) KYSE200 and KYSE450 cells were treated with 0 Gy (control) or 5 Gy X-ray, and CDC20 immunoprecipitates were subjected to Western blotting with the indicated antibodies.

**Figure 4 ijms-25-08383-f004:**
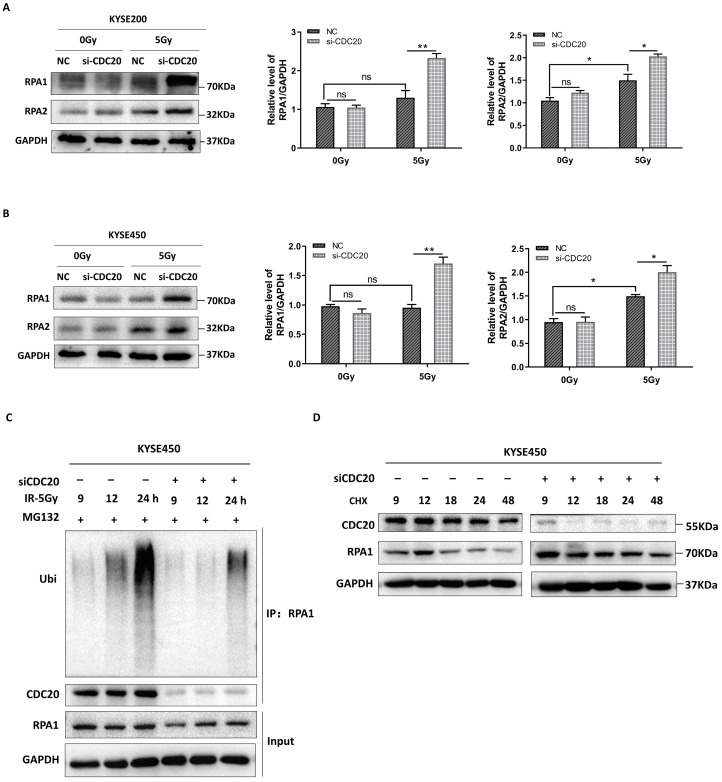
CDC20 regulates the ubiquitination of RPA1 protein. (**A**,**B**) KYSE200 (**A**) and KYSE450 (**B**) cells with siCDC20 or NC were exposed to 5 Gy X-ray, and the expression of RPA1 and RPA2 proteins was detected for 24 h. (**C**) KYSE450 cells transfected with indicated siCDC20 were treated with X-ray for 9, 12, and 24 h and with the indicated doses of MG132 (10 μM, Topscience, Shanghai, China) for 4 h before harvest. Signal intensities of the ubiquitinated RPA1 bands are shown below each blot. (**D**) CDC20 knockdown KYSE450 cells were treated with 50 µg/mL cycloheximide (CHX, Selleck, Shanghai, China), collected at different time points, and immunoblotted with antibodies against RPA1, CDC20, and GAPDH. RPA1 protein half-life is prolonged in CDC20 knockdown cells. GAPDH was used as an internal control. * *p* < 0.05, ** *p* < 0.01, ns = not significant.

**Figure 5 ijms-25-08383-f005:**
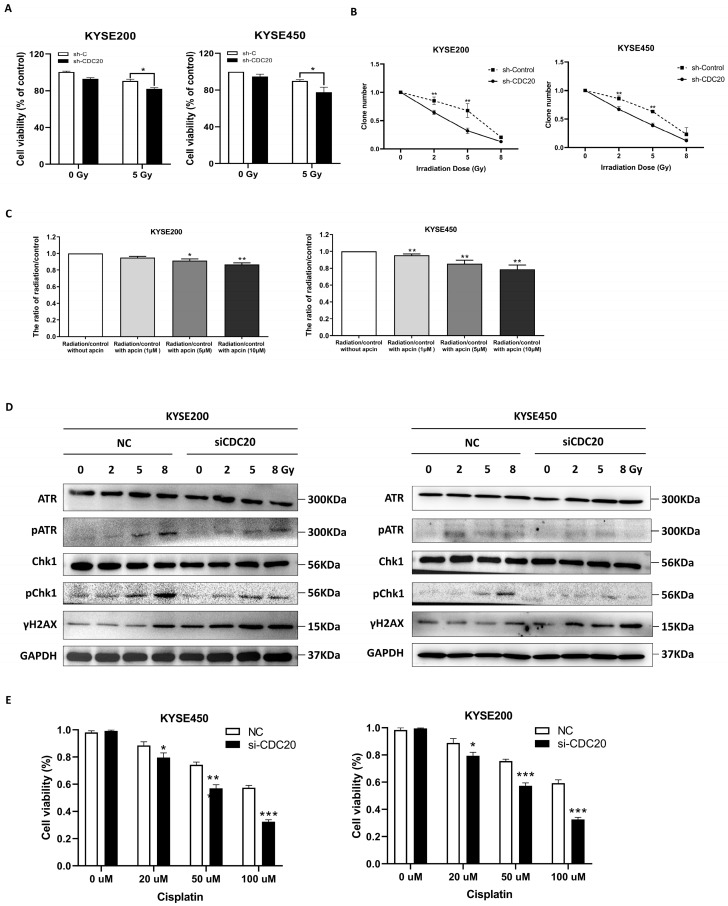
CDC20 regulates radio- and chemosensitivity in cancer cells. (**A**) KYSE200 and KYSE450 cells with reduced CDC20 expression and control vector cells were exposed to the 5 Gy X-ray, and cell viability was measured 24 h later. (**B**) A clonogenic assay of KYSE200 and KYSE450 cells with reduced CDC20 expression and control vector cells was carried out after irradiation with different doses of X-rays. (**C**) KYSE200 and KYSE450 cells were pretreated with different doses of apcin (0–10 µM) for 24 h before 5 Gy X-ray irradiation, and cell viability was detected 24 h after irradiation. (**D**) The expression levels of ATR, pATR, Chk1, p-Chk1, and γH2AX were detected by Western blotting in the CDC20 knockdown KYSE200 and KYSE450 cells compared with the NC cells upon the 5 Gy X-ray. (**E**,**F**) KYSE450 and KYSE200 cells with siCDC20 or NC were exposed to different doses of cisplatin (**E**) and etoposide (**F**), and cell viability was measured after 24 h. (**G**,**H**) The expression levels of p-Chk1 and γH2AX were detected in the CDC20 knockdown KYSE450 cells compared with the NC cells upon treatment with the indicated doses of etoposide (**G**) or cisplatin (**H**). GAPDH was used as an internal control. (**I**) KYSE450 cells were treated with different concentrations of cisplatin (**left**) or etoposide (**right**), and the protein levels of CDC20 and γH2AX were analyzed by Western blotting 24 h later. (**J**) KYSE450 cells with reduced CDC20 expression and control vector cells were treated with 20 μM cisplatin (**left**) and etoposide (**right**), and the expression level of γH2AX protein was detected 24 h later. Data were pooled from three independent experiments, and the results were represented as mean ± SD. * *p* < 0.05, ** *p* < 0.01, *** *p* < 0.001, ns = not significant.

**Figure 6 ijms-25-08383-f006:**
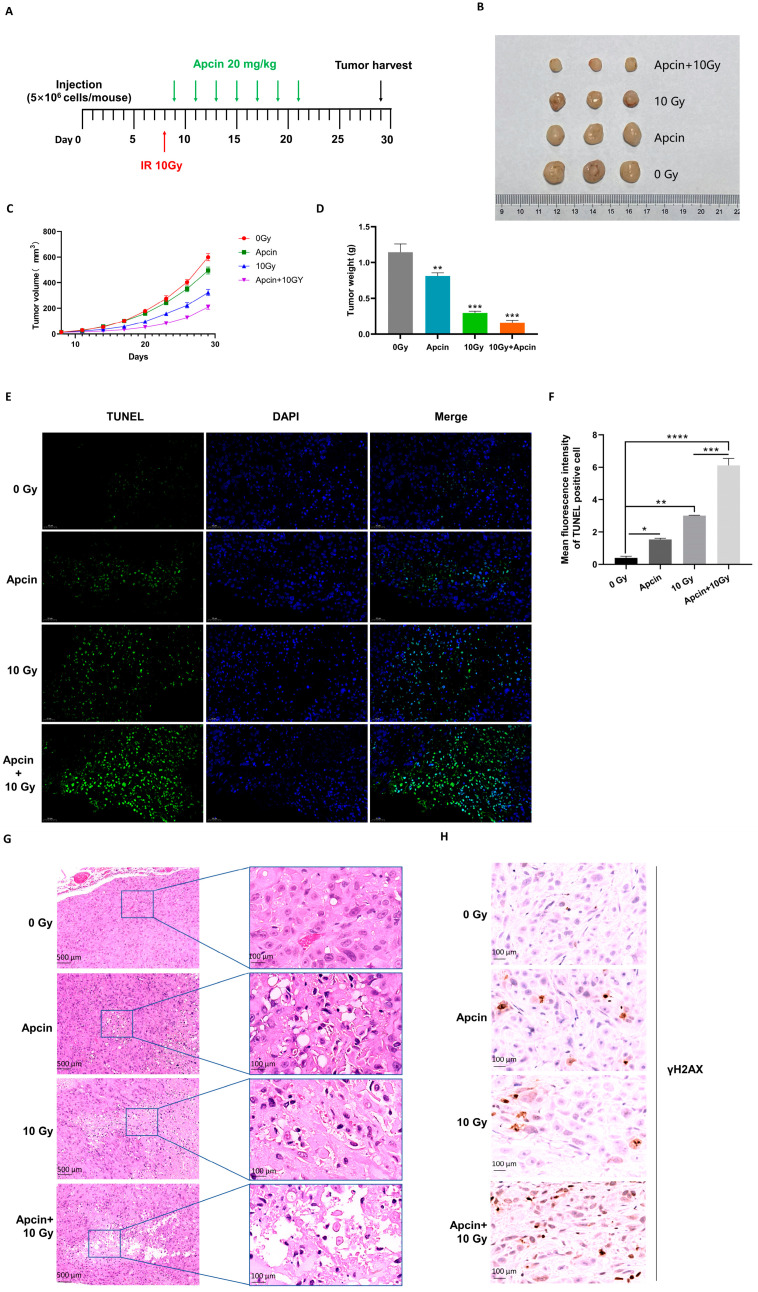
Apcin regulates radiosensitivity in vivo. (**A**) A schematic of the experimental schedule was shown. (**B**) Tumor xenografts of each group are shown. Each group of mice was composed of three BALB/c-nu/nu mice, and all nude mice were inoculated with KYSE450 cells (5 × 10^6^). On the eighth day after inoculation, the mice were irradiated with 0 and 10 Gy X-rays. On the ninth day of vaccination, Apcin (20 mg/kg) was injected intraperitoneally. On the 29th day after inoculation, the mice were sacrificed, and tumors were obtained. (**C**,**D**) The volumes (**C**) and net weights (**D**) of the tumors in each group are shown. (**E**) Apoptotic cells (green) in the tumor sections were identified by TUNEL assay, and the nuclei were counterstained with DAPI (blue). The scale bar represents 50 µm. (**F**) The number of apoptotic cells shown in (**F**) was quantified using GraphPad Prism. (**G**) Necrotic cells in tumor sections were identified by H&E staining. Representative images of tumor sections under different doses of X-ray and apcin treatment are shown. The scale bar represents 500 and 100 µm. (**H**) Immunohistochemical analysis of γH2AX expression in the tumor xenografts. The scale bar represents 100 µm. Data were pooled from three independent experiments, and the results were represented as mean ± SD. * *p* < 0.05, ** *p* < 0.01, *** *p* < 0.001, **** *p* < 0.0001.

## Data Availability

Data is contained within the article and Appendix A.

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
