# Peer review of "CDC20 Holds Novel Regulation Mechanism in RPA1 during Different Stages of DNA Damage to Induce Radio-Chemoresistance"

_ijms, 2024, doi:10.3390/ijms25158383_

Round 1
Reviewer 1 Report
Comments and Suggestions for Authors
The manuscript titled “CDC20 Holds Novel Regulation Mechanism in RPA1 during Different Stages of DNA Damage to Induce Radio-Chemoresistance” by Gao, Y.; et al. is a scientific work where the authors assessed the crosstalk driven by CDC20 to RPA1 which is relevant since CDC20 expression levels have been associated to poor prognosis and increased aggressiveness of certain cancers. The most relevant outcomes found by the authors could open new gates in the design of the next-generation of cancer therapies exploiting the molecular targets found in this research. The manuscript is generally well-written and this is a topic of growing interest.
However, it exists some points that need to be addressed (please, see them below detailed point-by-point) to improve the scientifc quality of the submitted manuscript paper before this article will be consider for its publication in the International Journal of Molecular Sciences.
1) KEYWORDS. The authors should consider to modify the term “radio-chemosensitivity” by “tumoral cell radio-chemosensitivity” in the keyword list.
2) INTRODUCTION. “The DNA damage response (…) tumor cells initiate DNA damage repair (…) in different types of cancer” (lines 31-38). Here, it may be convenient if the authors could provide quantitative data details about the worldwide global cancer burdens. This will significantly aid the potential readers to better understand the significance of this research.
3) The literature (…) DNA damage repair mechanisms (…) tumor cells (survival or death)” (lines 34-36). Here, even if I agree with this statement provided by the authors, it should be highlighted how cellular protein machinery processes can lead to DNA degradation [1] and how they can serve as molecular target against cancer diseases [2].
[1] Novo, N.; et al. Beyond a platform protein for the degradosome assembly: The Apoptosis-Inducing Factor as an efficient nuclease involved in chromatinolysis. PNAS Nexus 2022, 2, pgac312. https://doi.org/10.1093/pnasnexus/pgac312.
[2] Siklos, M.; et al. Therapeutic targeting of chromatin: status and opportunities. FEBS J. 2022, 289, 1276-1301. https://doi.org/10.1111/febs.15966.
4) “For instance (…) activation of PARP1 (…) ECT2 expression (…) carcinoma” (lines 38-41). Please, the authors should define the full-name of those terms which appear for the first time in the main manuscript body text. Then, the abbreviation should be placed between brackets. This comment should be taken into account for the rest of the main manuscript body text.
5) RESULTS. “2.1. Radiation induces nuclear accumulation of CDC20 and affects DNA damage response” (lines 73-113). Why did the authors not monitor larger radiation intensities in longer scale times? A brief statement should be provided in this regard.
6) Then, Figure 1 (line 102). The standard deviation bars should be added to the tested conditions in the control group before X-ray radiation exposure (0 Gy). Then, the lateral scale bar should be added in the panels related to the fluorescence microscopy measurements (C and F). Same comment for the Fig. 3, panel D (line 157) and Fig. 6, panels E and G (line 279).
7) DISCUSSION. This section clearly states the most relevant outcomes found in this research. No actions are requested from the authors.
8) CONCLUSIONS. Even if it is optional, it may be desirable to add some brief statements to highlight the significance of this work and also to outline some future action strategies to pursue this research.
9) MATERIALS & METHODS. “4.2. qRT-PCR analysis” (lines 408-418). Did the authors observe some issues related to the overspilling between the fluorescent channels during the data acquisition? Some information should be furnished in this regard.
Comments on the Quality of English LanguageThe manuscript is generally well-written albeit it may be desirable if the authors could recheck it in order to polish those final details susceptible to be improved.
Reviewer 2 Report
Comments and Suggestions for Authors
The topic sounds interesting, and the results are reliable. It is worth to be published. However, the manuscript was not well organized, and there still has some unclear points.
1) Please define all abbreviations at their first appearance to help the reader's understanding.
2) Regarding Fig.1: (B) The loading control for immunoblotting was GAPDH but not ACTB. (C) Please include scale bars in immunofluorescence images. Why are there no immunofluorescence images of KYSE70 and HCT116? (D, E) Quantify your immunoblotting data, and please ensure that cell line abbreviations are spelled correctly. (F) The following text in the manuscript "The results revealed that CDC20 knockdown cells exhibited significantly higher γH2AX levels compared to control cells post-IR" cannot be understood without the quantification of the number of γH2AX foci and the results of statistical analysis. Please include scale bars in immunofluorescence images. Overall, it is difficult to know how many hours have passed since radiation exposure, so please improve this.
3) Regarding Fig.2: Even though the following is written at line 84 "In response to the high incidence of esophageal cancer in the Chaoshan region of Guangdong Province, tumor cells, primarily KYSE200 and KYSE450, were selected for subsequent experiments. Additionally, HCT116 cells were also selected to investigate the potential role and mechanism of CDC20 in pan-cancer.", the cell lines used in the results are not consistent. This is very difficult to read, so we request that this be improved. (F) RAD51, which is involved in the HR repair pathway, was not significantly increased by radiation in the KYSE450 cell line. Isn't it difficult to understand the contribution of the HR repair pathway in this cell line?
4) Regarding Fig.3: Proteomic analysis was performed using the KYSE450 cell line, which makes it difficult to understand the contribution of HR repair pathways. Why was it evaluated in only one cell line?
5) Regarding Fig.4: The content on line 178 "CDC20 knockdown cells displayed increased level of RPA1 in the 0 Gy group" cannot be read from the graph.
6) Why did you choose eight cell lines?
7) Please state the source of purchase for major reagents and media.
8) Please provide more details about the radiation exposure conditions and LC-MS/MS analysis.
9) The statistical analysis method is described as follows "All data are presented as the mean ± standard error", but most figure legends read it as follows "the results were represented as mean ± SD".
Round 2
Reviewer 1 Report
Comments and Suggestions for Authors
The authors did a great deal of effort to cover the suggestions raised by the Reviewers. For this reason, the manuscript scientific quality was greatly improved. Based on the significance of this research, I warmly endorse_this work for further publication in IJMS.